# Pediatric Functional Neurological Symptoms Disorder: Walking Ability and Perceived Exertion Post-Pediatric Rehabilitation

**DOI:** 10.3390/ijerph20021631

**Published:** 2023-01-16

**Authors:** Jana Landa, Maya Gerner, Etzyona Eisenstein, Sharon Barak

**Affiliations:** 1Department of Pediatric Rehabilitation, The Edmond and Lily Safra Children’s Hospital, The Chaim Sheba Medical Center, Ramat Gan 5265601, Israel; 2The Sackler School of Medicine, Tel Aviv University, Tel Aviv 39040, Israel; 3Department of Nursing, Faculty of Health Science, Ariel University, Ariel 40700, Israel

**Keywords:** somatization, rehabilitation, gait, exertion

## Abstract

Background: Adolescents with functional neurological symptoms disorder (FNSD) commonly present walking abnormalities. Walking is influenced by ‘objective’ (e.g., fitness) and ‘subjective’ (e.g., fear) components. Rate of perceived exertion (RPE) reflects the interaction between these two components. This study compared the walking ability and RPE before and after rehabilitation of adolescents with FNSD to adolescents with moderate-to-severe traumatic brain injury (TBI). Factors predicting walking and RPE were also examined. Methods: Adolescents with FNSD (*n* = 31) and adolescents with moderate-to-severe TBI (*n* = 28) aged 6 to 18 years participated in the study. Participants received a multidisciplinary rehabilitation program. Six-minute walk test (6MWT) and RPE were assessed before and after rehabilitation. Results: At pre-test, the TBI group presented lower RPE than the FNSD group (3.38 ± 2.49 and 6.25 ± 2.71, respectively). In the FNSD group, pre-test 6MWT was a significant predictor of post-test 6MWT (adjusted R^2^ = 0.17; *p* = 0.01). In the TBI group, post-test 6MWT was significantly predicted by both the pre-test 6MWT and age (adjusted R^2^ = 0.16; *p* = 0.04). Conclusions: Prior to the intervention, adolescents with FNSD perceived walking as a more difficult activity than adolescents with TBI. Post-intervention, although the intervention was effective in terms of changes in 6MWT and RPE, the ‘subjective’ component still contributed to the elevated RPE of the FNSD group.

## 1. Introduction

Pediatric functional neurological symptoms disorder (FNSD) refers to impaired voluntary motor or sensory function without a recognized medical basis (e.g., a neurological basis). Recently, the fifth edition of the Diagnostic and Statistical Manual of Mental Disorders [1] (a guidebook widely used by mental health professionals in the diagnosis of many mental health conditions) introduced the term FNSD, assigning the disorder to the category of “somatic symptom and related disorders”. These disorders are mental health disorders characterized by intense physical (somatic) symptoms, yielding significant difficulties in performing the activities of daily living.

FNSD has a multifactorial etiology, encompassing ‘objective’ and ‘subjective’ components. In the ‘objective’ component, adolescents with FNSD present neurological changes (e.g., weakness) with neuro-imaging evidence/findings. For example, functional neuro-imaging research shows that, compared to healthy individuals, in response to painful stimulus, patients with FNSD present increased activity of the limbic regions [2]. In addition, anatomical structural imaging shows that patients with FNSD show a decrease in gray matter density. Taken together, these findings suggest that there is an altered emotional appraisal of stimuli in FNSD [2].

Another work [3] found that patients with FNSD presented increased functional connectivity in numerous brain areas, such as within and between the sensorimotor network, the default mode network, the salience network, and the dorsal attention network. Therefore, the authors concluded that individuals with FNSD have attention deficits that lead to external stimuli misperception. In addition, the attention deficits may result in failure to regulate bodily functions involved in interactions with external stimuli [3]. Finally, Conejero et al. [4] reported that in a cohort of patients with FNSD, resting brain metabolism and brain markers were associated with motor symptoms and recovery.

In terms of the ‘subjective’ component, adolescents with FNSD are clinically considered to be more sensitive than their peers [5]. For example, adolescents with FNSD present a wide range of psychological features related to attachment formation, post-trauma-related behavior, and abnormal expectations regarding the meaning of symptoms [6]. Another interesting aspect of FNSD is the lack of mentalization ability for painful emotions (e.g., frustration, anger), which leads to a “speaking body” [7]. More specifically, this approach views FNSD as a defense mechanism against unspeakable intra-psychic conflict/pain and a kind of ‘body language’ conveying distress [8]. It is important to understand the impact of the aforementioned ‘subjective’ component in FNSD, as it impacts health (e.g., diabetes mellitus, coronary artery disease, and asthma), physical symptoms (e.g., people who are depressed may suffer more if they become injured than non-depressed people), and physical disorders (e.g., physical functioning ability, such as walking ability) [9].

FNSD patients commonly present abnormalities in walking [10,11]. The ‘objective’ component of walking consists of physiological factors, such as sweating and heart rate (HR). The ‘subjective’ component of walking may consist in the motivational level and fear of the expected pain involved in movement. Among adolescents with FNSD, the ‘subjective’ and ‘objective’ components may interact [12], and this may be reflected in their walking ability. For example, during walking, physiological stimuli such as heart rate may be intensified because of a complex interplay with psychological stimuli such as pre-walking expectation, previous walking experience, and misdirected attention to walking strain in unconscious processes [13]. Moreover, environmental factors (e.g., parental factors) may also influence a child’s vulnerability and perception of physiological stimuli [14]. For example, parents’ perceptions regarding their child’s complaints, as well as characteristics of parent–child communication (e.g., the nature of conversation pertaining to the child’s symptoms) were found to be associated with the intensity of symptoms and functioning in adolescents with FNSD [11,14,15].

Rate of perceived exertion (RPE) may reflect the interaction between the ‘subjective’ and ‘objective’ components. RPE is the strain involved in physical exertion [16] (e.g., ‘not tired at all’ to ‘very, very tired’). RPE is given in response to the question of how ‘hard’ an exercise feels for an individual [17]. The objective component of RPE involves physiological changes during exercise (e.g., increased HR) [18]. The subjective component involves psychological factors (e.g., mood states) [19]. The subjective components may have an impact on interpretation of the ‘objective’ (i.e., physiological responses) sensations arising from the body during physical exertion and may also influence walking ability.

Numerous reviews have been conducted on the topic of RPE [18,20,21]. However, these reviews focused on adults, while the usage of RPE and its effect on walking ability in adolescents with FNSD have not been established. It is important to examine RPE among adolescents with FNSD, as this may shed light on the interaction between the ‘subjective’ and ‘objective’ components affecting walking ability. Understanding walking perception among adolescents with FNSD will help to construct appropriate gait interventions for patients. Moreover, over the years, groups of adolescents with FNSD with movement impairment have been mostly compared with groups with orthopedic injuries or healthy controls [22,23]. However, as mentioned, neuro-imaging shows that the FNSD group is characterized by changes in brain anatomy and function [24,25,26]. Therefore, it is appropriate to compare FNSD recovery to adolescents with neuroanatomy changes, such as traumatic brain injury (TBI). A comparison group of adolescents with a known organic explanation for the walking impairment will aid in understanding the walking ability of adolescents with FNSD and the relative importance of the ‘subjective’ components. More specifically, if adolescents with FNSD present an RPE higher than that of adolescents with TBI, then this group of adolescents is a unique group in the rehabilitation setting and special attention should be given to their perception of walking and the ‘subjective’ components.

FNSD is a common diagnosis in neurology practice [27], often resulting in various comorbidities [28,29], impairing the patients’ quality of life, and resulting in high health care use and costs [6]. This intensifies the need to develop better knowledge of functional ability in patients with FNSD [30]. Studying gait recovery is of special importance, considering the high prevalence of such deficiencies among adolescents with FNSD [10,11]. However, function in patients with FNSD is influenced by the interplay between physiological stimuli (e.g., heart rate, sweating rate, ventilation rate) and psychological aspects (e.g., pain- and strain-related anxiety) [31]. Therefore, in order to better understand gait ability among adolescents with FNSD, aspects pertaining to both the body’s physiological responses and the interpretation of the physiological responses must be studied jointly. Heart rate and RPE measures during walking can aid in understanding the interplay between the physiological (‘objective’) and psychological (‘subjective’) aspects of walking. Therefore, this study aimed to better understand gait recovery and RPE while walking among adolescents with FNSD. More specifically, the study aims were (1) to compare the walking ability and RPE of adolescents with FNSD before and after rehabilitation to the walking ability and RPE of adolescents with moderate-to-severe traumatic brain injury (TBI) receiving a standardized rehabilitation program; and (2) to examine factors predicting discharge walking ability and RPE. Identifying differences in the walking/RPE recovery of both groups may contribute to understanding the recovery process and the usage of RPE scales in FNSD, thus further improving rehabilitation programs. It was hypothesized that both study groups would increase their walking distance. However, in comparison to FNSD, the changes in walking distance in TBI would be greater, as FNSD walking distance might also be influenced by a more complex psychological component through gait impairment. In addition, it was hypothesized that TBI patients RPE would not improve to the same extent as the FNSD group.

## 2. Materials and Methods

### 2.1. Study Participants

Study participants were adolescents with FNSD (research group) and TBI (control group). Data were collected retrospectively from medical files between 2017 and 2019.

#### 2.1.1. FNSD Group (Research Group)

Inclusion criteria: (1) age: 6-to-18 years; (2) FNSD diagnosis based on the DSM-V (Conversion Disorder-300.11; F44.4; [1]); (3) treated at the ambulatory clinic; and (4) presenting lower limb weakness, gait disorders, and/or sensation or paralysis disorder due to FNSD [1]. More specifically, gait disorder diagnosis was based on the observations of a licensed physical therapist during indirect (i.e., informal behavioral observations) and direct (the 6MWT) evaluations. Based on Daum et al. [32], positive clinical signs in gait disorders such as dragging monoplegic gait, excessive slowness, constant falls, large amplitude body sway, walking with the knee flexed, or complete inability to walk were documented. Forty percent of participants could not walk or used walking aids (walker or crutches); 60% of participants were able to walk but demonstrated an abnormal gait pattern.

Exclusion criteria: (1) pain with no sensory and/or motor symptoms/complaints; (2) exhibition of only FNSD upper-extremity symptoms; (3) treated in inpatient rehabilitation; and (4) other clinical diagnoses, such as other psychiatric diseases and/or physical impairments.

#### 2.1.2. TBI Group (Control Group)

The control group was composed of adolescents with TBI from across the country referred to the inpatient rehabilitation center in the center of the country.

Inclusion criteria: (1) age: 6 to 18 years; and (2) moderate-to-severe TBI, resulting in motor impairments (e.g., hemiparesis).

Exclusion criteria: (1) exhibition of only upper-extremity disorders; and (2) other clinical diagnoses.

A total of 31 patients with FNSD (mean age: 13.61 ± 2.79) and *n* = 28 patients with TBI (mean age: 12.37 ± 3.31) met the inclusion criteria of the study. The prevalence of females in the FNSD group was statistically significantly greater than in the TBI group (67.7% and 32.1%, respectively; chi-squared = 754; *p* = 0.02). For additional demographic and clinical characteristics of the study participants, refer to Table 1.

### 2.2. Procedures

This is a historic prospective study [33]. Study procedures were approved by the Institutional Review Board of Tel Hashomer Hospital (7394–20-SMC). All adolescents (TBI and FNSD groups) received the full conventional multi-modal rehabilitation program. A description of the study procedures for each group follows.

#### 2.2.1. FNSD Group

The FNSD group received an integrative pediatric rehabilitation program, which consisted of treatment by a multidisciplinary team including a physician, occupational therapy, physiotherapy, individual educational lessons, weekly family-team meetings (once a week or fortnight), and psychological therapy. The program’s goals were minimizing symptom recurrence (e.g., movement/walking impairment) and return to age-appropriate functioning. The main discharge criterion was the return to age-appropriate activity, such as school attendance and age appropriate independence in activities of daily living. The program was conducted over 1–2 weeks in an ambulatory format. The following is a brief description of both the psychological and physical therapy treatments.

Psychological therapy consisted of sessions with adolescents and with parents. The sessions with adolescents mainly aimed at (1) enhancing the feeling of agency and establishing mind–body unity, through psychoeducation; and (2) encouraging adolescents to verbally share “negative” feelings (frustration, fear, etc.) with parents and, in parallel, express difficulties and set rehabilitative goals with the physical team. The sessions with parents aimed at (1) understanding FNSD mind–body unity; (2) psychoeducation; (3) identifying current parent–child communication and creating novel direct communication patterns with the child (e.g., conflict acknowledgment facilitation); and (4) providing practical help in managing adolescents’ hardships (e.g., in school).

Physical therapy consisted of individual and group therapy. Each session lasted 45 minutes. The key principles of the physical therapy sessions were (1) therapist–child rapport—in order to enhance therapist–child rapport, the therapist demonstrates trust in the child’s pain and disability, listens to the child’s complaints, and presents understanding of and empathy for the child’s difficulties and needs; (2) functional goals—in order to achieve the functional goals prescribed, the following principles were applied. (a) The child was asked to also practice the specific target function at home, with each patient prescribed one to three exercises to perform at home. All exercises were adapted to individual ability, for example, laying bridging (i.e., raising hips off the bed/floor), standing/sitting weight shifting, and functional strength exercises specifically designed for the involved (i.e., weak) limb (e.g., kicking a ball, standing up from stool). The number of repetitions was progressively increased and discussed with the patient during the weekly meetings. It is also important to note that prescribed home exercises were an integral part of the program for which the child was responsible and was required to report progress [30]. (b) The child was empowered to create and set his/her own functional goals (e.g., climbing stairs). (c) The child must display constant progression (even minor) in the chosen target [34]. (d) Educational concepts such as psychoeducation were introduced. For example, the importance of verbalizing and expressing discomfort and difficulty in words rather than in bodily avoidance. In addition, in therapy, the body scheme concept was introduced, meaning that difficulty in one bodily part affects the entire body’s function. (e) Treatment emphasizes function and not a specific body part, for instance, focusing on walking and not on gait pattern [35]. As such, physical therapy sessions comprised many functional exercises and home training. The sessions included at least one session at the gym to train functional strength, cardiovascular fitness, and flexibility. No special modalities (e.g., electrical stimulation) were incorporated.

For further detailed information about the program’s components, see Gerner et al. [8].

#### 2.2.2. TBI Group

The TBI group also received an integrative pediatric rehabilitation program. However, unlike the FNSD group, treatment was highly influenced by the child’s cognitive function.

Psychological therapy sessions with adolescents and with their parents focused on: (1) the medical traumatic event; (2) the expected outcomes following the injury and; (3) psychoeducation and emotional support for the parents. Psychological therapy was provided once to twice a week.

Physical therapy: children with TBI present numerous physical impairments, such as altered muscle tone, proprioception, and balance. Such impairments commonly limit the ability to independently perform age-appropriate activities and instrumental activities of daily living [36], as well as participation. Therefore, physical therapy should commence as soon as possible, once the child is clinically stable [36,37]. Physical therapy commonly involves the following types of therapy: preventing secondary complications (e.g., contractures and weakness), sensory stimulation, fitness, and functional training (e.g., sit-to-stand training and gait training [38]. Physical therapy was conducted at least twice a day, six days a week. Each physical therapy session lasted 45 minutes. Both individual and group therapy were provided. Physiotherapists commonly conducted functional treatments, such as gait education and bed mobility. Physical agents and other modalities (e.g., hydrotherapy, electrotherapy, and cryotherapy) were also used.

### 2.3. Outcome Measures

#### 2.3.1. Demographic and Clinical Characteristics

The demographic (age and sex) and clinical (age at injury, injury chronicity, and hospitalization duration) characteristics of both the FNSD and TBI groups were collected retrospectively from medical files between 2017 and 2019. Somatization severity was only assessed for the FNSD group with the Child’s Somatization Inventory-24 (CSI-24) [39]. The CSI-24 has both a child self-report form and a parent proxy-report form. The child’s questionnaire consists of a list of 24 items concerning symptoms experienced by the child within the past two weeks. Each item is rated for its frequency on a five-point scale (0 = not at all; 4 = a whole lot). Number of symptoms (0–24) and symptom intensity were calculated separately, with higher scores indicating higher somatic complaint intensity. The parent form is identical to the child form, except that the questions are worded differently. In the current study, the child’s form was used (children’s questionnaire Cronbach’s alpha = 0.87) [39].

#### 2.3.2. Hemodynamic Characteristics

Resting, exercising, and recovery HR were measured using a Polar watch S810 combined with a chest strap. Resting HR was established while sitting after 10 minutes rest. Mean 6MWT HR was calculated. Participants’ predicted maximal HR was calculated using Tanaka’s maximal HR equation (208 − 0.7 × age) [40]. Participants’ exercising HR was converted to percentage of maximal HR. Exercising at 64 to 76% of maximal HR was considered moderate-intensity physical activity [41]. Recovery HR was measured in sitting for one minute after completion of the 6MWT.

#### 2.3.3. Rate of Perceived Exertion

RPE was evaluated at the end of the 6MWT using the OMNI Walk/Run RPE (OMNI-RPE). The scale consists of four pictures of a child (boy or a girl) walking up a hill and progressively looking more tired (e.g., leaning forwards). The OMNI-RPE is valid for use within typically developing children/adolescents ages 8 to 18 years [42,43,44]. According to Roemmich et al. [44], children exercising at approximately 60 and 70% of maximal HR report OMNI scores of 1.82 ± 0.3 and 3.85 ± 0.4.

#### 2.3.4. 6MWT

6MWT was used for assessing functional walking ability [45]. Participant’s walking ability relative to the norm was evaluated by calculating the number of standard deviations (SDs) below the mean for the appropriate age group, according to the norms published by Geiger et al. [46].

### 2.4. Statistical Analysis

Demographic and clinical characteristics are presented using descriptive statistics. Between-group (research vs. control group) differences in demographic and clinical characteristics were evaluated using an independent *t*-rest (continuous variables) or the chi-square test (categorical variables).

Changes between pre- and post-intervention (T0 and T1, respectively) in all study variables were examined via paired t-tests. 6MWT distances and SDs below the norm at T0 and T1 were also presented graphically using box plots. Changes from T0 to T1 were examined via Cohen’s d effect size (ES) (mean ∆/SD average of two means) [47]. A correction for the dependence among means was conducted using Morris and DeShon’s equation [48]. In ES, values <0.20, 0.20–0.50, 0.51–0.80, and >0.80 are considered trivial, small, moderate, and large ESs [47]. Between-group differences at T0 and T1were also evaluated using independent t-tests. One sample t-tests were used in order to compare the current sample’s RPE to those observed at similar percentages of maximal HR in Roemmich et al.’s study [44]. Variables associated with T2 6MWT walking distance and RPE-OMNI were evaluated using Pearson correlation coefficients. Finally, two separate forward multiple stepwise regression analyses for factors predicting 6MWT distance and RPE-OMNI at T2 were conducted. The purpose of this was to examine how variables obtained during admission to rehabilitation can predict discharge walking ability and RPE. Only variables significantly correlated with the aforementioned two variables were included in the regression model. Multicollinearity examination was conducted for all independent variables, using the variance of inflation factor >10 [49]. The criterion for inclusion in the model was an alpha level of 0.05, while for exclusion an alpha level of 0.10 was used.

## 3. Results

### 3.1. Changes from Pre- to Post-Test: Within and between Differences

#### 3.1.1. FNSD Group

From pre- to post-test, statistically significant changes were observed in all outcomes evaluated, except for exercising HR. More specifically, in terms of walking ability, participants statistically significantly increased their 6MWT walking distance (pre-test: 280 ± 120; post-test: 400 ± 125; ES = 0.55; Figure 1) and decreased 6MWT standard deviations from the norm (pre-test: 6 ± 2.5; post-test: 5 ± 2.6; ES = −0.58; Figure 2). Regarding hemodynamic characteristics, participants statistically significantly decreased their resting (pre-test: 82.28 ± 14.68; post-test: 77.07 ± 11.92; ES = −0.35) and recovery HRs (pre-test: 98.41 ± 15.94; post-test: 92.00 ± 18.50; ES = −0.59). Finally, RPE statistically significantly decreased (pre-test: 6.25 ± 2.71; post-test: 4.25 ± 2.61; ES = −0.73).

At pre- and post-tests, participants’ exercising HR was approximately 60% and the OMNI score was 6.25 and 4.25, respectively. According to one sample *t*-test, the FNSD group OMNI at both the pre- and post-test was statistically significantly greater than that of apparently healthy children (*p* < 0.001).

#### 3.1.2. TBI Group

The TBI group presented statistically significant changes from pre- to post-test only in walking ability. More specifically, this group statistically significantly increased their 6MWT distance (pre-test: 282 ± 150; post-test: 450 ± 100; ES = 1.20; Figure 1) and decreased their 6MWT standard deviations from the norm (pre-test: 6.5 ± 2.1; post-test: 3.01 ± 2.2; ES = −1.18; Figure 2).

At pre- and post-test, participants’ exercising HR was approximately 70% and OMNI scores were 3.38 ± 2.49 and 3.97 ± 2.85, respectively. In a sample of apparently healthy children, at approximately 70% of maximal HR, the OMNI score was 3.85 ± 0.4. According to one sample t-test, no statistical between-group differences in OMNI score were observed between adolescents with TBI and apparently health children (*p* > 0.050).

#### 3.1.3. FNSD and TBI: Between-Group Differences

At pre-test, in comparison to the TBI group, the FNSD group presented statistically significantly lower exercising HR (FNSD = 61.17 ± 11.58; TBI = 69.40 ± 7.56; *t*-statistic = 5.34, *p* = 0.006) and statistically significantly higher RPE (FNSD = 6.25 ± 2.71; TBI = 3.38 ± 2.49; *t*-statistic = 8.04, *p* = 0.001). At post-test, in comparison to the TBI group, the FNSD group presented lower exercising HRs (FNSD = 59.30 ± 10.68; TBI = 68.54 ± 10.09; *t* = 11.40, *p* < 0.0001) and similar RPE (FNSD = 4.25 ± 2.61; TBI = 3.97 ± 2.85; *t* = 0.09, *p* = 0.90). Regarding walking ability, no between-group differences were observed pre-test. However, at post-test, the FNSD walking ability was better than that of the TBI group (Figure 1 and Figure 2).

### 3.2. Associations with and Prediction of Post-Test Walking Ability and RPE

#### 3.2.1. Post-Test Walking Ability

In the FNSD group, the post-test walking ability was statistically significantly correlated only with the pre-test walking ability (r = 0.449, *p* < 0.05; Table 2). The pre-test walking ability was also a significant predictor of post-test walking ability (adjusted R^2^ = 0.17; F = 6.32; *p* = 0.01; Table 3). In the TBI group, post-test walking ability was statistically significantly correlated with both pre-test walking ability and age. The aforementioned variables also significantly predicted post-test walking ability (adjusted R^2^ = 0.16; F = 1.08; *p* = 0.04; Table 2 and Table 3).

#### 3.2.2. Post-Test RPE

In both study groups, only pre-test RPE was statistically correlated with and predicted post-test RPE (Table 2 and Table 3) (FNSD: adjusted R^2^ = 0.22, F = 7.52, *p* = 0.001; TBI: adjusted R^2^ = 0.53, F = 20.16, *p* = 0.0004).

## 4. Discussion

The perception of strain involved in physical activity (RPE) is significantly associated with physical measures of exertion (e.g., HR and oxygen consumption) [16]. RPE is influenced by both objective and subjective components. Therefore, the ability to detect strain might be altered among adolescents with FNSD. To date, there are no data regarding adolescents with FNSD’s walking RPE and possible changes following treatment. Therefore, the current study aimed to bridge the two aforementioned domains (‘objective’ and ‘subjective’ components of walking) by studying adolescents with FNSD’s walking ability using their RPE and comparing their responses to adolescents with TBI, an organic impairment in the central nervous system.

At its simplest level, RPE is influenced by physiological factors, such as muscle fatigue [50,51]. However, modern models [52,53] show that subjective scores of RPE are also thought to be influenced by an often overlooked aspect, namely, ‘affective’ or emotional qualities [51,54,55], such as exertion and anxiety sensations. In the current study, although the physiological stress in adolescents with FNSD (i.e., percentage of maximal HR) was statistically significantly lower than that of adolescents with TBI, the RPE of adolescents with FNSD was statistically significantly higher or similar. These results may suggest that in comparison to adolescents with TBI, among adolescents with FNSD, the affective, and not the physiological, component has the greater impact on RPE.

The impact of the affective aspects on adolescents with FNSD was not surprising, as the prevalence of emotional conditions (e.g., depression and anxiety) in this population is >4 times higher than in the non-FNSD population [56]. Moreover, depression and anxiety are often accompanied by cognitive-affective disturbances. It has been stated that neural correlates of cognitive-affective amplifiers are integrated into a neurocircuit framework for somatosensory processing. Cognitive-affective disturbances, including appraisal (negative anticipation), attentional bias, pain catastrophizing, negative emotion, and alexithymia, enhance the effects on visceral somatic processing [57]. An interesting support to the aforementioned study has emerged from Kanbara et al. [58], whose findings show a dissociation between subjective and objective responses in patients with somatization. The authors suggest that patients with somatization have an altered awareness of feelings arising from the body, thus supporting the concept of “alexisomia” or “escaped bodily feelings” in patients with somatization. Similarly, the Bayesian model states that somatic symptoms are induced and maintained by prior experience of body-focused attention, symptom expectations, and beliefs about illness. Somatic symptoms are much less induced by environmental cues and sensory information [59]. Therefore, these adolescents have a high anticipation of pain. The current study’s results, along with those of the aforementioned previous studies, demonstrate a major difference between the gait rehabilitation for adolescents with TBI and FNSD. While gait training for adolescents with TBI focuses on improving fitness and gait pattern, among adolescents with FNSD, great emphasis must be placed on introducing educational concepts (i.e., psychoeducation), for example, the importance of verbalizing and expressing discomfort and difficulty in words rather than in bodily avoidance.

At post-test, although RPE was still relatively high compared to the physiological stress level, RPE had considerably decreased in the FNSD group. In order to understand this reduction in RPE, it is important to evaluate changes observed in cardiovascular fitness. From pre- to post-test, the FNSD group significantly increased their 6MWT distance, with no change in the percentage of maximal HR. These results may indicate an improvement in cardiovascular fitness. However, it is reasonable to assume that the decrease in RPE was more related to affective changes and not to physiological (fitness) changes. Namely, the percentage of maximal HR was similar at both the pre- and post-tests and, therefore, RPE should not have significantly decreased simply because of improvements in fitness. Therefore, the decrease in RPE may be more related to body awareness changes that occurred during treatment. Changes in body awareness are of special importance, as patients with FNSD are vulnerable individuals. Their vulnerability to developing abnormal beliefs regarding movement difficulties is based on predisposing factors such as past illness experiences, illness beliefs, and cognitive and affective biases. The aforementioned factors may have a pivotal role in a predisposition to augment salient sensory information arising from the body during physical exertion. Vulnerability is also a potent factor in conditioning learning and could facilitate another key proposed element of the generation of FNSD: abnormal self-directed attention [60].

Individuals differ greatly in the extent to which they pay attention to their body, i.e., body awareness. Body awareness training, thought to increase visceral awareness, is associated with greater emotional response coherence [61]. In order to achieve greater body awareness, during both psychological and physical therapy, adolescents were taught and trained in verbalizing their emotional awareness, and mind–body unity was established through psychoeducation and enhancing feelings of agency. More specifically, during physical therapy, educational concepts (i.e., psychoeducation) were introduced. For example, participants were encouraged to verbalize the soma (e.g., pay attention to sensations of fatigue based on breathing rate). This is also consistent with mindfulness-based treatments encouraging nonreactive attention to sensations arising from the body, which thus reduce negative interpretations of these sensations [62,63,64]. Such treatment strategies may, not only teach children to listen and identify their actual body sensations, but may also target the autonomic system. Dysregulation of the autonomic nervous system in adolescents with FNSD has been documented and therefore is an important component of therapy [65].

Thus, efforts should be made to try to engage adolescents in physical therapy. Several strategies have been suggested to increase the collaboration of adolescents with FNSD in therapy. For example, expression of trust regarding the child’s complaints, addressing ambivalence regarding treatment effectiveness, and taking strategies to make the adolescent a full member in the program (e.g., the rate of progression should be coordinated with the adolescents) [8]. There are several limitations to this study. First, important physiological factors such as maximal oxygen consumption were not measured. Second, there is also a need to include other self-reported outcomes pertaining to participants’ mental-health status. This might further aid our understanding of RPE and the personal experience of children/adolescents with FNSD during exertion. Third, although the ESs were large, the sample size was small, therefore limiting the generalizability of the study and the ability to make more certain conclusions regarding the contribution of an integrative rehabilitation approach to RPE and walking ability in children/adolescents with FNSD. Finally, in the current study, there was no zero control group. We did not include such a control group, as it is recommended that adolescents with FNSD receive rehabilitation services and, considering the impact of FNSD on quality of life, denying or prolonging reception of treatment might be considered unethical. It is important to note that on average children with FNSD in the study started rehabilitation 4.37 + 3.25 months from initiation of symptoms with no improvement or deterioration in their condition. Therefore, it is less likely that the improvement in function occurred solely on account of natural recovery and it is reasonable to assume that the changes observed were indeed influenced by the treatment received.

## 5. Conclusions

Adolescents with FNSD perceive exertion (RPE) differently from adolescents with TBI. More specifically, the perception of exertion among adolescents with FNSD is considerably greater than that of adolescents with TBI, especially at the beginning of rehabilitation. Post-intervention, FNSD patients’ RPE was still high relative to their percentage of maximal HR. As RPE is influenced by both ‘subjective’ (e.g., anxiety pertaining to the activity) and ‘objective’ (e.g., HR) components, it appears that the subjective components of RPE play a greater role in walking among adolescents with FNSD compared to adolescents with TBI. These results stress the importance of integrating strategies such as psychoeducation into physical therapy for adolescents with FNSD, along with exercises to improve fitness and gait pattern.

## Figures and Tables

**Figure 1 ijerph-20-01631-f001:**
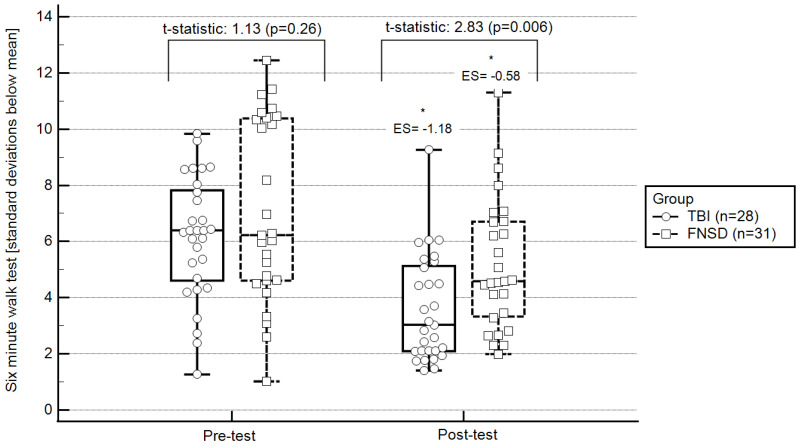
Pre- and post-test walking ability relative to the norm: within- and between-group differences. Notes: * statistically significant within-group changes (*p* < 0.05; 2-tailed); ES, effect size of within-group changes from pre- to post-test (ES is based on a single pooled standard deviation (Cohen’s d = mean ∆/standard deviation_average from two means_) and was corrected for dependence between means). FNSD, functional neurological symptoms disorder; TBI, traumatic brain injury; central box,- values from the lower to upper quartile (25 to 75 percentile); vertical line, minimum to the maximum values; middle line, median.

**Figure 2 ijerph-20-01631-f002:**
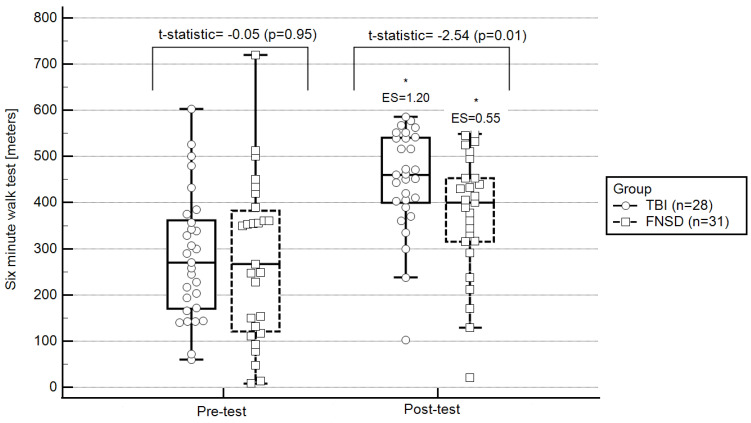
Pre- and post-test walking ability: within- and between-group differences. Notes: * statistically significant within-group changes (*p* < 0.05; 2-tailed); ES, effect size of within-group changes from pre-to post-test (ES is based on a single pooled standard deviation (Cohen’s d = mean ∆/standard deviation_average from two means_) and was corrected for dependence between means); FNSD, functional neurological symptoms disorder; TBI, traumatic brain injury; central box, values from the lower to upper quartile (25 to 75 percentile); vertical line, minimum to the maximum values; middle line, median.

**Table 1 ijerph-20-01631-t001:** Demographic and clinical characteristics of study participants.

Variables			Traumatic Brain Injury (*n* = 28)	Functional Neurological Symptoms Disorder (*n* = 31)	Between-Groups *t*-Statistic OR Chi-Squared (*p* Value)
			Man (SD) OR *n* (%)	Mean (SD) OR *n* (%)	
Demographic characteristics	Age, years: mean (SD)		12.37 (3.31)	13.61 (2.79)	1.76 (0.17)
	Sex: *n* (%)	Males	19 (67.9) ^b^	10 (32.3) ^a^	7.54 (0.02)
		Females	9 (32.1) ^b^	21 (67.7) ^a^	
Clinical characteristics	Age at injury, years: mean (SD)		12.20 (3.51)	-	N/A
	Chronicity at pre-test, months: mean (SD)		2.37 (1.85)	4.37 (3.25)	1.75 (0.16)
	Total hospitalization time, months: mean (SD)		7.64 (5.02)	6.26 (3.64)	0.49 (0.61)
	Hospitalization type: *n* (%)	Inpatient	10 (35.7) ^b^	0 (0) ^a^	53.12 (<0.0001)
		Outpatient	3 (10.7) ^b^	31 (100) ^a^	
		Inpatient + outpatient	15 (53.6) ^b^	0 (0) ^a^	
	Children’s Somatization Inventory, number of symptoms: mean (SD)		-	14.23 (5.11)	N/A
	Children’s Somatization Inventory, severity: mean (SD)		-	28.18 (14.48)	N/A

Notes: ^a^, statistically significantly different than “Traumatic brain injury” (*p* < 0.05; 2-tailed); ^b^, statistically significantly different than “Functional neurological symptoms disorder”; SD, standard deviation.

**Table 2 ijerph-20-01631-t002:** Associations with walking ability and rate of perceived exertion post-test.

Variables	Traumatic Brain Injury (*n* = 28)		Functional Neurological Symptoms Disorder (*n* = 31)	
	Post-Test Six-Minute Walk Test, Meters	Post-Test, OMNI Score	Post-Test Six-Minute Walk Test, Meters	Post-Test, OMNI Score
Age, years	0.332 *	−0.117	0.049	−0.388
Total hospitalization time, months	−0.028	−0.183	0.105	0.270
Pre-test Six minute Walk Test, meters	0.333 *	−0.139	0.449 *	−0.243
Pre-test Six-minute Walk test OMNI, score	0.211	0.747 *	−0.159	0.505 *
Pre-test resting heart rate, beats/minute	0.158	0.066	−0.180	0.290
Pre-test exercising heart rate, percentage from the maximum	0.125	−0.271	0.011	0.199
Pre-test recovery heart rate, beats/minute	0.145	0.145	−0.168	0.249

Notes: * Statistically significant correlations at a *p* < 0.05 level (two-tailed).

**Table 3 ijerph-20-01631-t003:** Variables predicting walking ability and exercising rate of perceived exertion (OMNI) post-test.

Dependent Variable	Study Group	Independent Variables	Coefficient	Standard Error	*t*	*p*	Variance Inflation Factor
Post-test 6MWT	Traumatic brain injury (*n* = 28)	Constant	290.78				
		T1—6MWT	0.23	0.14	0.65	0.04	1.00
		Age	12.68	4.85	2.96	0.04	1.001
		Adjusted R^2^ = 0.16; F = 10.08 (*p* = 0.04)					
	Functional Neurological Symptoms Disorder (*n* = 31)	Constant	280.08				
		T1-6MWT	0.33	0.13	2.51	0.01	1.00
		Adjusted R^2^ = 0.17; F = 6.32 (*p* = 0.01)					
Post-test-OMNI	Traumatic brain injury (*n* = 28)	Constant	0.67				
		T1-OMNI	0.69	0.15	4.49	0.004	1.00
		Adjusted R^2^ = 0.53; F = 20.16 (*p* = 0.0004)					
	Functional Neurological Symptoms Disorder (*n* = 31)	Constant	1.20				
		T1-OMNI	0.48	0.17	2.74	0.01	1.00
		Adjusted R^2^ = 0.22; F = 7.52 (*p* = 0.001)					

Note: 6MWT, Six-minute Walk Test; only variables significantly correlated with the dependent variable were included; in models in which more than one variable was included in the analysis, variables were entered in order of correlation strength.

## Data Availability

Data will be available upon request.

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
