# Peer review of "Pediatric Functional Neurological Symptoms Disorder: Walking Ability and Perceived Exertion Post-Pediatric Rehabilitation"

_ijerph, 2023, doi:10.3390/ijerph20021631_

Round 1

Reviewer 1 Report

Thank you for the opportunity to review this manuscript. The purpose of the study was to investigate the influence of a rehabilitation program on walking ability and the rate of perceived exertion in patients with functional neurological symptoms disorders and in patients with traumatic brain injury..
I can see the interest in the topic for the readership of the scope of IJERPH, although there are some major points that arose during reading, along with some methodological flaws that need to be revised before the manuscript can be considered for publication.

English grammar has to be checked thoroughly and in-depth. I recommend native speaker revision by an institution.

Abstract/Title:

General:

The abstract needs to be revised in terms of writing clarity, content summarization, and clear elaboration of the study’s highlights. L16-17 needs clear revision as it is not clear what you’re addressing. As an objective you highlight the comparison to a group of patients with traumatic brain injury: why do you compare your findings to ‘apparently healthy children’ in L20/21?

Manuscript:

Introduction:

General: Try to elaborate the theoretical background better, so that an interested reader can follow you better on the unfolding of the research area, research gap, and adjunct research questions.

It is not fully clear, why you actually did this investigation!

L30 et seq.: The whole introduction requires grammatical and spelling revision.

L33: What does that mean?

L37 et seq.: Be more aware of correct referencing in theoretical background content.

L39: Referencing required. What does that contribute? What does this brain damage lead to?

L45: Referencing required.

L49 et seq.: I do not fully understand what these facts contribute or explain to your topic. Be more aware of an audience that is not totally into the topic of neurological disorders and pathologies.

L68 and L73: Just because something hasn’t been done so far is not a very powerful argument for research. Aren’t there important unsolved questions in the field of neurological disorders that can be addressed by your research? And as you write in L73, it ‘might be appropriate’ – Why might this be good?? After disentangling the theoretical background and elaborating a research question, you should be convinced by investigating something, right?

Methods:

General:

Your study lacks a zero-control group. Without the zero-control group, there is no possibility to directly refer to potential motor improvements due to the applied rehabilitation protocol. With the zero-control group, you would have important information, on how this generally influenced/improved motor skills, such as walking. Without the controls, I could assume that any improvements are at odds due to the intervention program itself. It doesn’t help to answer your research question from my point of view.

L89: Sentence appears incomplete.

L94: How did you standardize gait disorders? Did you define any thresholds towards step width and length, for example, or how did you exactly build up your groups? More information is mandatory.

L109: I have never read this term of study definition.

L132-165: Could you specify the training volume quantities? This would be an important point of your publication, to give details about the rehabilitation program you applied. Or in general, tell us the total training volume, what exercises have been performed, and so on. For example, L159-165 describes the general necessity of training in TBI children, which is undoubtedly relevant, but you should still have any metrics about training bouts. There might be physicians out there, who would also apply the program to achieve enhancements in patients with similar symptoms.

Results:

L225-229: English grammar!

Table 1: Please revise the Table according to clearer editing. Needs quite some improvement.

Table 1: In the row “age injury…” you provide data only for the brain injury group, which is fine, but you do have not any data for the FNSD group, but nonetheless you provide some statistical metrics for between-group analysis. How did you calculate statistical metrics if the one variable is zero or N/A?

L235 et seq.: It would be good to have also some absolute values in the Results section or relative changes from pre to post-testing, not just effect sizes.

Based on the presented Results and the aforementioned flaws concerning the study design without any zero controls, you should consider a modification of the outline and objective of your paper towards that what you actually did: A comparison of two impaired groups while receiving a standardized rehabilitation program.

Figure 1 et seq.: You need to work out your Figures, as well, more meticulous, in terms of editing. In the Y-axes units need to be depicted with [], the figure has no title…Figures and Tables need to fulfill stand-alone criteria and need to be understood by any reader without reading any part of the manuscript.

L261-264: Try to place these few lines before the Figures. OR place the Figure at the end of the section, otherwise, a reader could easily miss this information.

Discussion:

General:

-        I stumble repetitively about the term ‘youth’. I suggest using a clearer term such as ‘adolescents’ or something else.

-        I miss the clear discussion of your results to the results you presented from other research. You rather structured your Discussion in a way that you, again, repeat what’s already known in the field and then how your Results need to be interpreted. You should merge these findings together to give us some connections/links to the research that found similar results to you and the research that is rather contradictory to your results. Out of the discussion, you should deduce your conclusion and practical implication(s).

L315-318: English grammar!

L321: Digits 12 and lower need to be written out.

L337-340: Sentence structure and length!

L373 et seq.: You should modify this section and include that did not have any zero-controls in your study. That would also a point to discuss in the Discussion, because this has important impact on the generalizability and practical implication(s) of your results and findings.

Conclusion:

I have to apologize, but I do not follow or get your conclusions, especially the reasoning for the changed perceived RPE. Could you transfer these findings and their interpretation less hedged, please? This appears as another repetition of the Results. But here you need to give clear conclusive sentences about your research. Be aware some readers might just read your conclusion to be encouraged to read your whole paper.

Thanks for your work. See all points as an encouragement to enhance your manuscript that should actually meet some important quality levels before it actually can be scientifically published.

Author Response

Thank you for the comments. Attcahed are my point by point responses.

Reviewer 2 Report

Review comments:

This manuscript entitled “Pediatric Functional Neurological Symptoms Disorder Walking Ability and Perceived Exertion post-paediatric rehabilitation” aimed to portray walking ability and RPE among youth with FNSD and TBI at the beginning and at the end of rehabilitation; and examine factors predicting discharge walking ability and RPE. The authors identify differences in walking recovery/RPE of both groups may contribute to understanding recovery process and the usage of RPE scales in FNSD and thus to further improve rehabilitation programs.

Although this study was read with interest, more work needs to be done in the manuscript to show clearly the potential of the study. In the Introduction Section, the author should emphasize the role of TBI in the study. It is unclear to understand the relationship between FNSD and TBI. In the Methodology Section, analytical method used should be emphasized that how to examine factors predicting discharge walking ability and RPE. The written content of the research conclusion is inconsistent with the purposes of the research. It is also suggested that conclusions should complement the practical application of research findings.

Author Response

Thank you for the comments. Attached are my point by point responses. 

Reviewer 3 Report

Congratulations to the authors of this publication.

Topics important from the point of view of the quality of life, physiotherapy, psychotherapy of children with FNSD and TBI. The original solution to the problems posed. Groups properly selected, numerous for inclusion and exclusion criteria, properly designed.

My only remark concerns table 1. In my opinion, it should be included in the description of the research material.

After a minor correction, the work is, in my opinion, ready for publication.

Author Response

(The authors gave the same response as above.)
